

# Comparison of the Performance between Three Doppler wind Lidars and a Novel Wind Speed Correction Algorithm

Yidan Zhang[1], Hancheng Hu[2], Mengqi Liu[1], Fugui Zhang[1], Huilian She[1], Hao Wu[1, *]

[1] Key Laboratory of Atmospheric Sounding, College of Electronic Engineering, Chengdu University of Information Technology, Chengdu, 610225, China

[2] College of Optoelectronic Engineering, Chengdu University of Information Technology, Chengdu, 610225, China

*Correspondence to*: Yidan Zhang (3230305012@stu.cuit.edu.cn)

**Abstract.** Doppler wind Lidars (DWLs) have been widely used to detect wind vector variations, based on ground monitoring of atmospheric boundary layer and wind shear. This study evaluates the performance between three DWLs and in situ balloon radiosonde. Lidars data comparison focus on the low altitudes (height < 2 km) from July to September 2021 from three producers: MSD (Minshida), CUIT (homemade), and WP (windprofile) Lidars. Comparison of results shows the root mean square errors (RMSE) for wind speed were 1.11 m/s, 4.45 m/s, and 5.15 m/s, while wind direction RMSE were shown at 49.83°, 82.89°, and 84.87°, respectively. The measurement accuracy decreases with the altitude increase (<2km). Particle mass concentration loading has positive correlation on the Lidar performance, when $PM_{2.5}$ ranges from 35 to 50 µg/m³, MSD Lidar exhibited the highest wind speed correlation ($R^2$ = 0.82) with radiosonde, and the wind direction accuracy observed by the three Lidars is enhanced with the increase of aerosol concentration, indicating that particle loading is the critical factor affecting the wind profile. Lidar performance varied significantly with planetary boundary layer heights (PBLH), three Lidars demonstrated optimal performance at lower altitudes (500-750m), with the Pearson correlation coefficients (PCCs) of wind speed are 0.97, 0.92, and 0.72, while the wind direction is shown at 0.98, 0.75, and 0.70, respectively. The vertical relationship between cloud base height (CBH) and PBLH had also varied influences on the Lidar measurements. Machine learning was used to remove anomalies and complement the missing values, the random forest (RF) demonstrated superior performance with the Area Under the Curve (AUC) of 0.93(CUIT) and 0.90(WP) in the Receiver Operating Characteristic (ROC) curves. RF-based correction of CUIT enhanced $R^2$ from 0.42 to 0.65. The $R^2$ between RF-based CUIT and Aeolus satellite is shown as 0.83, indicating that the method effectively improved data, even in circumstances of anomalies. We proposed a new correction algorithm combined with the isolation forest (IF) and RF to handle high-dimensional and incomplete datasets. Our procedure could increase the Lidar measurement quality of the wind.



## Introduction

The development of the low-altitude economy depends on efficient airspace management and flight scheduling. The Lidar
technology has laid a strong foundation for turbulence measurement, wind shear detection, gravity wave analysis, and
boundary layer height estimation (Chanin et al., 1989; Harvey et al., 2015; Sathe and Mann, 2013; Shun and Chan, 2008;
Talianu et al., 2006). The biggest, most significant risk of unmanned aerial vehicle (UAV) flight is the wind shear in the low
layer at the boundary. DWL uses the optical Doppler effect to measure atmospheric wind speed by detecting the frequency
shift between emitted and backscattered laser signals, offering high spatial and temporal resolution measurements(Du et al.,
35 2017).

Conventional wind measurement systems face inherent limitations. In recent years, Lidar has successfully overcome many of
the limitations associated with conventional detection equipment (Liu et al., 2019). For example, differing from mechanical
anemometers, DWL can remotely measure wind speed without contact with the atmosphere (Tavakol Sadrabadi and Innocente,
2024). Radiosondes, reckoned as the best accuracy, suffer from discontinuous temporal sampling and cannot support all-
weather monitoring (Abdunabiev et al., 2024). In observational experiments, there are phenomena leading to anomalies and
missing DWL data. These errors may arise from different atmospheric conditions, for instance, the strong aerosol concentration
and Brillouin backscattering signals may lead to errors in retrieving low-altitude wind speeds (Shen Fahua et al., 2021).
Traditional Lidar data inversion methods (e.g., Velocity Azimuth Display, VAD; Doppler Beam Swinging, DBS) exhibit
horizontal wind speed errors exceeding 10% in complex terrains (Liu et al., 2022). Differences in pulsed laser instruments can
affect the detection efficiency and accuracy of Lidar's detection (Ge et al., 2014), as well as data processing methods (Smalikho
and Banakh, 2016).

Machine learning has been demonstrated to have the ability to solve missing values and improve DWL accuracy, such as noise
filtering and data imputation (Lin et al., 2022; Lolli, 2023; Yang et al., 2021). Meteorological data have the characteristics of
time series, and machine learning methodologies such as the RF and neural networks have been proved effective in unveiling
latent patterns in wind-related time series data. The incorporation of machine learning-based validation and quality control
algorithms has the potential to enhance wind measurement accuracy and facilitate the prediction of upper-level wind fields. In
recent years, wind field data has received a lot of attention, and the RF algorithms are particularly popular (Vassallo et al.,
2020; Wang et al., 2017). For example, the RF algorithm has been used to correct numerical model wind predictions for
weather forecasting (Wang et al., 2021), improving forecast accuracy significantly. The RF algorithm employs an ensemble
of decision trees to mitigate overfitting and enhance prediction robustness (Hastie et al., 2009). It has been proved to enhance
prediction accuracy without a substantial increase in computational cost, to be robust against multicollinearity, and to
demonstrate considerable stability in scenarios involving anomalies (Boulesteix et al., 2011). The RF algorithm has been
demonstrated to address missing data effectively and to manage high variability, rendering it well-suited for the preprocessing
of wind datasets (Zhao et al., 2024b). In comparison to other algorithms, such as AdaBoost and K-nearest neighbors (KNN),
RF demonstrates superior performance in predicting wind speed and power generation, as evidenced by reduced mean absolute





percentage error (MAPE) values (Malakouti, 2023). This study proposes the RF algorithm for Lidar wind data to develop a wind profile correction algorithm. For the verification of wind profiles, a radiosonde will be used to enhance the stability of the system and evaluate the feasibility of the algorithm (Huang et al., 2021).

Spaceborne wind Lidar technology is also effective for wind detection (Kim et al., 2021). Satellite retrieval for wind field information has become an important trend for future applications. The combination of ground-based and spaceborne Lidar enables high-precision atmospheric wind speed observation, which is crucial for weather forecast and wind energy development (Evgenia et al., 2021; Rennie et al., 2021). Satellites equipped with scatterometers and radiometers, such as Metop-A, Metop-B, and Coriolis (He et al., 2022a; Silva et al., 2022), provide wind speed and direction. The Aeolus satellite, launched by the European Space Agency in 2018, is the first to provide comprehensive global wind observation. It operates in a 320 km sun-synchronous orbit, following a flight path roughly along the Earth's day-night boundary, and completing one orbit every 90 minutes (Evgenia et al., 2021). The satellite provides high-quality wind components and aerosol optical properties from the Earth's surface to the lower stratosphere(Evgenia et al., 2021; Flament et al., 2021). The satellite with a 1.5 meter diameter Lidar system emits ultraviolet laser pulses and collects scattered light particles from the atmosphere at altitudes of 20-30 km. Wind speed, direction, and other parameters are determined by measuring the Doppler shift of the light waves (Witschas et al., 2020). This technology is one of the most effective measurements. In 2021, Guo et al. compared data from the European Space Agency's satellite with domestic wind profiler RWP network measurements, finding a good match between the Aeolus wind product and the RWP data. Chen et al. examined the seasonal variation in Aeolus satellite detection performance in China by combining ERA5 and radiosonde data, concluding that the satellite's performance is influenced by seasonal factors (Siying et al., 2021). However, there is still a gap between comparing and validating Aeolus satellite products and Lidar data. Joint comparisons of spaceborne and ground-based measurements are essential for assessing the advantages and limitations of Lidar in accurately capturing wind fields, which will support the integration of laser sensors and inversion algorithms in next-generation wind measurement satellites.

This study investigates wind field measurements using three ground-based Doppler Lidar systems (CUIT, MSD, and WP Lidars) through a three-month comparative campaign at the Nanjiao Observatory in Beijing, collocated with radiosonde observations. the accuracy of the three ground-based Lidars is evaluated against radiosonde data as the reference standard. The study investigates the impact of $PM_{2.5}$ concentration on wind measurement performance. The effect of height on wind speed and direction was analyzed by comparing the Lidar performance under different PBLH and CBH conditions. The study also conducts satellite-ground validation to assess the consistency of the Aeolus Satellite. We propose a novel machine learning framework for wind profile correction by comparing various algorithms to optimize the data accuracy.



## 2 Instruments and methods

### 2.1 method and instruments

The experiment was conducted at Beijing's Nanjiao Observatory (39.80°N, 116.32°E, 30 m a.s.l.) from June 9 to August 31, 2021, featuring a three-month intensive comparative observation campaign with multiple Lidar wind measurement systems. The Nanjiao Observatory, an integrated atmospheric observation base of the China Meteorological Administration. It plays a significant role in monitoring and predicting weather changes in the Beijing region. The observatory stands as the sole upper-air meteorological station within a 200-kilometer radius, and launches enhanced radiosondes every day at 01:15, 07:15, and 19:15 LST. As the radiosondes ascend with the ballon, they drift with the wind and collect upper-air wind field data. These balloons can climb to at least 40 km altitudes, providing wind field data within the region.

As shown in Fig. 1, Three coherent DWLs—MSD (Minshida Technology Co.), CUIT (homemade), and WP (WindPrint S4000)—were deployed alongside daily radiosonde launches. Both the MSD Lidar and CUIT Lidar employ single-frequency pulsed fiber lasers with a wavelength of 1550 nm. Aerosol molecules and large particles present in the air serve as tracers of the wind field. Coherent DWL retrieves the atmospheric wind field by measuring the backscatter of aerosols moving with the wind field(Weickmann et al., 2009).

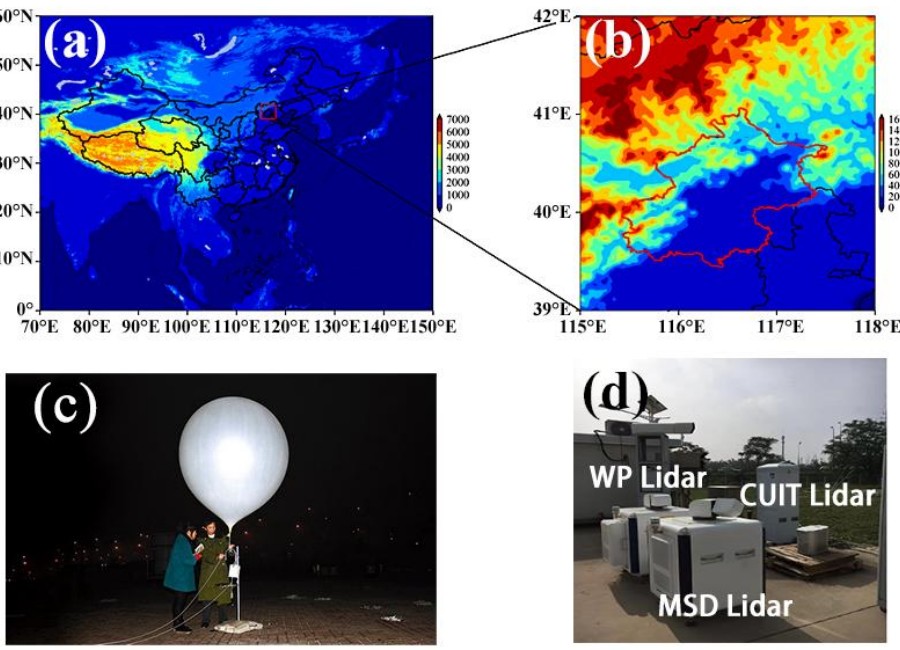

**Fig. 1. Location and instruments of the radiosonde station.**





This study necessitates retrieving Aeolus satellite data from the European Space Agency (ESA) website (https://aeolus.services/) for comparative analysis. Aeolus is a wind-profiling satellite, launched by ESA in 2018. It operates in a 320 km Sun-synchronous orbit, following a flight path approximately aligned with Earth's day-night terminator, completing an orbit every 90 minutes (Evgenia et al., 2021). The satellite is equipped with a 1.5 m diameter telescope, a scattering receiver to collect reflected signals, and a Doppler wind ultraviolet Lidar system named "Aladin," which operates with an output power comparable to a small nuclear reactor and can penetrate the atmosphere up to 30 km altitude. Its working principle involves a processing system with a 1.5 m diameter aperture emitting pulsed ultraviolet laser beams (wavelength 355 nm) at a rate of 50 observations per second, with each beam generating billions of photons directed at the atmosphere. However, only a few hundred are scattered back to the satellite due to interactions with atmospheric molecules. The Doppler effect determines the time delay between emitted pulses and backscattered signals. The Doppler effect determines the time delay between emitted pulses and backscattered signals, and the wind field is observed by calculating the wind direction, speed, and displacement. The mean wind speed measurements are obtained by averaging the values obtained in vertical and horizontal directions. Vertical sampling is conducted within 24 altitude bins, ranging from 0.25 km to 2 km.

A comparison of the technical specifications of Aeolus and other Lidars is presented in Table 1. The three Lidars use range gates to select specific distance ranges, measuring the velocity of aerosol particles within these ranges, and obtaining wind speeds at different altitudes. The Aeolus Level 2B (L2B) product is the Aeolus satellite's primary wind field product. It provides horizontal line-of-sight (HLOS) wind speed observations that have been atmospheric corrected and geo-located, extracting the necessary L2B data variables, such as the latitude, longitude, and wind speed information within the observation time range.

The L2B product also provides scene classification based on the backscatter ratios corresponding to winds from "cloudy" or "clear" atmospheric regions, generating observation types such as "Rayleigh-clear," "Rayleigh-cloudy," and "Mie-cloudy"(Borne et al., 2024; Martin et al., 2021).

**Table. 1. Instruments Technical Index.**

| Index | CUIT Lidar | MSD Lidar | Aeolus-Level 2B |
|---|---|---|---|
| Lidar detection Height | 50m~1500m | 50m~5000m | 0～30km |
| Range resolution | 50m | 30m/60m | 0.25-2km |
| Speed measurement range | 0~60m/s | -55m/s~55m/s | -150m/s-150m/s |
| Accuracy of speed measurement | ≤0.2m/s | ≤0.5m/s | 1～3m/s |
| Laser pulse band | 1550nm | 1550nm | 355nm |
| Power consumption | 76W | 300W~1000W | 850W |



| Size | 440mm×400mm×260mm | 700mm×700mm×1300mm | 4.6m×1.9m×2.0m |
|---|---|---|---|
| Weight | 21.5kg | 130kg | 1260kg |
| Working temperature range | -30℃~+50℃ | -30℃~+50℃ | |

## 2.2 Data processing

The collected data often have outliers in the DWL measurement process. Data use quality control can obtain accurate information about the changes in the atmospheric wind profile, which is helpful to understand and predict the atmospheric motion pattern. As shown in Fig. 2, the wind speed and direction data from the radiosonde, CUIT Lidar, MSD Lidar, and WP Lidar are height-matched through the implementation of the sliding window method, producing a comprehensive dataset that is arranged sequentially based on time, altitude, wind direction, and wind speed. This dataset serves as the foundation for subsequent analyses. The sliding window method, widely used in signal processing and time series analysis, was applied to align datasets. This method involves strategically restricting the maximum number of data points that each window can accommodate, as previously outlined in the extant literature (Wang et al., 2023; Zhao et al., 2024a). The specific matching process, using the radiosonde height and CUIT Lidar height as an example, employs a sliding window of size 3, moving two positions to the right each time. In each window, a value is selected and compared with the CUIT Lidar height, with the closest value being selected as the matched radiosonde height.

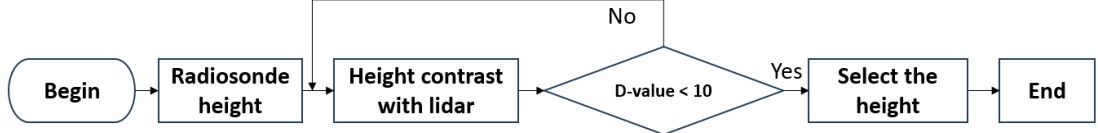

**Fig. 2. The flow of Lidar and radiosonde dataset height matching.**

When measuring wind speed, sudden peaks often result in anomalous values. The IF model is used to filter the data to identify and remove these anomalies. IF (Liu et al., 2008) is an unsupervised anomaly detection algorithm that effectively identifies anomalies in a dataset by isolating outliers(Hernandez-Mejia et al., 2024). The algorithm recursively partition data points into subsets using randomly selected features and thresholds (Borne et al., 2024). Anomalies require fewer partitions to isolate them from other data (Liu and Aldrich, 2024). Anomalous values have the characteristic of being few and significantly different from normal values. The IF can separate and remove anomalies without modeling the normal data, and identify anomalous data accurately. In constructing the binary tree structure, fewer partitions are required to isolate anomalous data, which is closer to the root, and normal data is further from the root. This feature allows for effective anomaly detection.





### 1.3.1 Isolation tree

Let $T$ be a node of the isolation tree (iTree). $T$ has two possibilities: it is either a leaf node with no children or an internal node with a test and exactly two children. $(T_l, T_r)$. The test is composed of an attribute $q$ and split value $p$, where $p > q$, which divides the data points into $T_l$ and $T_r$.

The sample data $X = \{x_1, x_2, \ldots, x_n\}$where $n$ represents the number of instances in the distribution. The iTree is recursively constructed by splitting based on the attribute $q$ and split value $p$. The splitting process terminates when the tree reaches the height limit ($|X| = 1$), or when all instances in $X$ have the same value.

### 1.3.2 Anomaly Detection

For anomaly detection, the method primarily ranks data points based on path length or anomaly score, with the points at the top being considered anomalous.

Path length: The path length $h(x)$of point $x$ is calculated as the distance from the root node of the iTree to the leaf node for point $x$.

Anomaly score:

After getting the path length $h(x)$, the outlier scour of $x$ is as follows:

$$S(x, u) = 2^{-\frac{E(h)}{C(u)}}, \tag{1}$$

$$C(u) = 2H(u-1) - \frac{2(u-1)}{u}, \tag{2}$$

Where the $u$ is the number of samples, and $C(u)$ is the average path length of all data in the training set. $H(i)$ is harmonic number,$\ln(i) + 0.5772156649$. $E(h)$is the average path length of $x$ across n iTrees.

(1)  When $E(h) \to C(u), S \to 0.5$;

(2)  when $E(h) \to 0, S \to 1$;

(3)  when $E(h) \to u - 1, S \to 0$;

Evaluate and remove outliers based on the anomaly score.

(1)  If $S(x, u)$ approaches 0.5, the outlier becomes less apparent.

(2)  If $S(x, u)$ approaches 0, the score is normal value.

(3)  If $S(x, u)$ approaches 1, the value is anomalous.

### 2.3 Random Forest for Lidar Data

To address the missing values after Lidar detection and after outlier removal, this study uses RF to correct the Lidar data. In the correction of Lidar data, the wind speed and direction at each altitude layer are treated as samples. Considering the




uncertainties and errors in the original data, the RF is used for correction. By integrating multiple decision trees, RF can effectively handle and analyze high-dimensional complex data, accurately predicting wind speed and direction, thereby improving wind field data's supplementation and prediction capabilities. It is important to note that the performance of the RF model largely depends on the quality of the training data and the selection of features. Additionally, attention should be given

to the issue of overfitting, and the model should be optimized and adjusted based on actual conditions. The RF model in this research is built as follows:

Step 1: Extract a sub-sample matrix from the training matrix as the training samples.

Step 2: Each sample has M features. Specify a constant m where $m \ll M$ and randomly select a subset of m features from the M features. Finally, select the optimal feature subset for regression.

Step 3: Allow the tree to continuously split until a certain height is reached.

Step 4: Repeat the previous three steps until the regression tree is fully constructed and trained. The final output model is the "ensemble predictor" $f(x)$. The ensemble predictor $f(x)$. is composed of the "base learners" $h_1(x), \dots, h_J(x)$ (Cutler et al., 2012):

$$f(x) = \frac{1}{J}\sum_{j=1}^{J} h_J(x), \tag{3}$$

### 2.4 Lidar data correction

The overall data processing workflow is shown in Fig. 3. After matching the wind direction and wind speed data from the three Lidars with radiosonde data using the sliding window method, the data is compared under different pollution conditions, PBLH, and weather conditions to identify the optimal performing Lidar and the Lidar that requires improvement. For missing

values caused by the instrument itself or following anomaly cleaning, cubic spline interpolation (CSI), back propagation neural network (BPNN), Genetic Algorithm (GA), k-nearest neighbor (KNN), and RF were used to fill the missing values. By comparing the correlation of each algorithm, the most suitable algorithm is identified for the final Lidar data optimization, the Aeolus satellite is used to verify the reliability of the algorithm further.





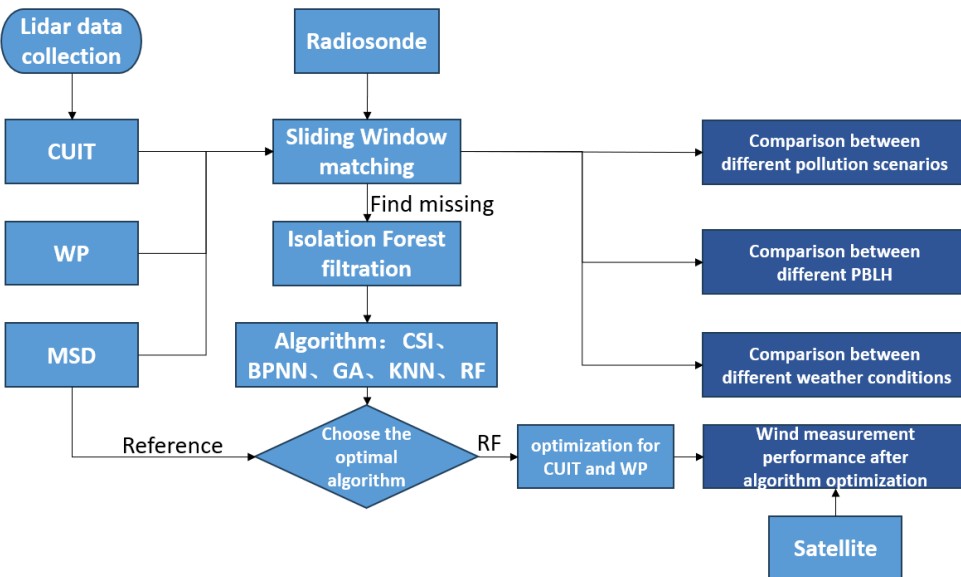

Fig. 3. The total flow of data processing.

## 2.    Results and discussion

### 3.1 Performance Comparison of Doppler Wind Lidars

Fig. 4 compares wind speed and direction data from MSD Lidar, CUIT Lidar, and WP Lidar with the radiosonde data at different heights. The dispersion of the scatter points represents the correlation between the Lidars and the radiosonde data. The WD was defined as the range from 350° to 10°, and all winds within this range were classified as northerly, which were not considered anomalies in the study. As shown in Fig. 4, MSD Lidar, CUIT Lidar, and WP Lidar exhibited good consistency with the radiosonde data in the low-altitude region below 600m. However, as the altitude increased, the dispersion of wind speed and direction from the three Lidars gradually increased, especially above 1500m. Regression parameters of three Lidars and radiosonde data are summarized in Table 2. MSD Lidar had a wind speed and direction slope of 0.99 and 0.81, respectively, with RMSE values of 1.11m/s and 49.83°, which were closest to the radiosonde data. CUIT Lidar showed significant anomalies below 750m and above 1500m, with wind speed overestimated, and wind direction RMSE reaching 82.89°. The performance of WP Lidar exhibited an overestimation of wind speed across all heights. Notably, the magnitude of errors was particularly pronounced in high-altitude regions, as evidenced by wind direction RMSE reaching 84.87°. Overall, the observation data in the low-altitude region (blue) were more stable. In contrast, the high-altitude region (red) decreased observation accuracy for all three Lidars due to altitude effects. This reveals an exponential decay trend in Lidar measurement accuracy with increasing altitude, consistent with the attenuation characteristics of Lidar backscatter signals. These results provide critical insights for



high-precision wind field monitoring: The MSD Lidar is the preferred choice for boundary layer observations (<1.5km). At the same time, real-time radiosonde data correction is advised for elevated altitude applications.

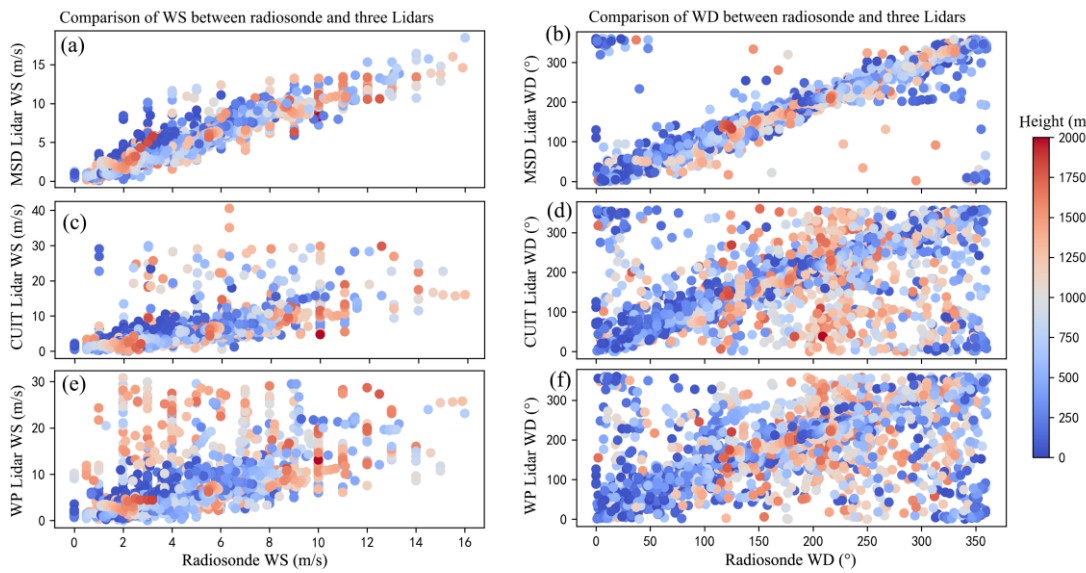

**Fig 4.** (a) WS and (b) WD of radiosonde data and MSD Lidar at different heights; (c) WS and (d) WD of WP Lidar; Comparison of (e) WS and (f) WD of CUIT Lidar; color bar represents height.

**Table. 2. Regression parameters of three Lidars and radiosonde.**

| Lidar | Condition | Fitted curve | N | RMSE |
|-------|-----------|--------------|-----|-------|
| MSD | WS | y=0.99x+0.51 | 2885 | 1.11 |
| | WD | y=0.81x+36.65 | 2885 | 49.83 |
| CUIT | WS | y=1.18x+0.67 | 2514 | 4.45 |
| | WD | y=0.54x+86.3 | 2885 | 82.89 |
| WP | WS | y=1.14x+1.69 | 2848 | 5.15 |
| | WD | y=0.48x+99.67 | 2885 | 84.87 |

## 3.2 Comparative analysis of performance under different air quality conditions

To investigate the performance of three DWLs in measuring wind speed and direction under different aerosol mass concentrations, the experiment integrated $PM_{2.5}$ concentration data from collaborative observations with wind profile analysis.



The PM$_{2.5}$ concentrations are relatively low at the site, so the concentration range was divided into three pollution levels: L1 (PM$_{2.5}$ = 0–15 µg/m³), L2 (PM$_{2.5}$ = 15–35 µg/m³), and L3 (PM$_{2.5}$ = 35–50 µg/m³). Fig. 5a – 5c present Scatter plots of wind

speed regression relationships for the MSD, CUIT, and WP Lidars across these pollution tiers, with linear regression lines for L1(red), L2(green), and L3(yellow). The results show the correlation of the three Lidars in different pollution levels. It is evident that aerosol concentrations significantly affect the performance of Lidar in wind speed detection. The regression parameters of the three Lidars and radiosonde under different pollution conditions are summarized in Table 3. During L3 pollution episodes, MSD Lidar achieves the highest correlation with the radiosonde ($R^2 = 0.82$), demonstrating strong

stability and reliability. In contrast, CUIT Lidar and WP Lidar show much lower correlations under L3 conditions ($R^2_{CUIT} = 0.24, R^2_{WP} = 0.04$), indicating that their detection performance is significantly affected by air quality. Under L1 conditions, the correlations for CUIT Lidar and WP Lidar are $R^2 = 0.35$  and  $R^2 = 0.32$, with RMSE values of 1.43 m/s and 1.36 m/s, respectively. Under L2 conditions, the correlations decrease to  $R^2 = 0.3$  and  $R^2 = 0.17$, with RMSE values of 1.45 m/s and 1.39 m/s, respectively. Aerosol mass concentration has a negative impact on the detection performance of DWL, particularly

for CUIT and WP Lidars, which exhibit significant performance degradation under higher pollution levels.

The wind direction difference can be used to evaluate the impact of different aerosol mass concentrations on the performance of DWL. Due to the periodic nature of wind direction data, the absolute value of the wind direction difference was calculated, and differences exceeding 180° were excluded from the analysis. Fig. 5d – 5f illustrates the distribution of wind direction differences as a function of PM$_{2.5}$ concentration and height. The x-axis represents aerosol mass concentration, the y-axis

represents height, and the colorbar represents the wind direction difference. The MSD Lidar exhibits high detection accuracy, maintaining wind direction deviations within 20°. Under the L1 air quality, maximum deviations (D>20°) occur below 400m altitude, while within the 400-1400m range, deviations remain below 10°. For L2 conditions within this height band, deviations increase to 17.5°. The wind direction difference of MSD Lidar remains below 7.5° at altitudes above 1 km, indicating high accuracy in high-altitude detection, though with certain limitations under low aerosol concentration conditions. The CUIT

Lidar demonstrates a heightened wind direction difference of 40°-65° when PM$_{2.5}$ concentrations fall below 17 µg/m³. When PM$_{2.5}$ concentrations increase to 17-37 µg/m³, the deviation reduces to 30°-40°. As PM$_{2.5}$ concentrations increase (>40 µg/m³), the difference significantly decreases to 10°-25°, indicating improved accuracy under higher pollution conditions. The WP Lidar demonstrates the poorest performance (<15 µg/m³ PM$_{2.5}$), with deviations reaching 50°-80°. However, when PM$_{2.5}$ concentrations exceed 40 µg/m³ and altitudes exceed 800m, deviations significantly reduce to about 10°. WP's performance

above 800m improves with deviations within 20° under L3 conditions. The observed accuracy enhancement with increasing aerosol concentrations (>800m altitude) likely stems from amplified laser backscattering signals caused by atmospheric particulates. This phenomenon particularly improves wind field retrieval accuracy in elevated regions. Operational deployment of DWL systems in polluted environments requires careful consideration of both instrument specifications and ambient aerosol characteristics.



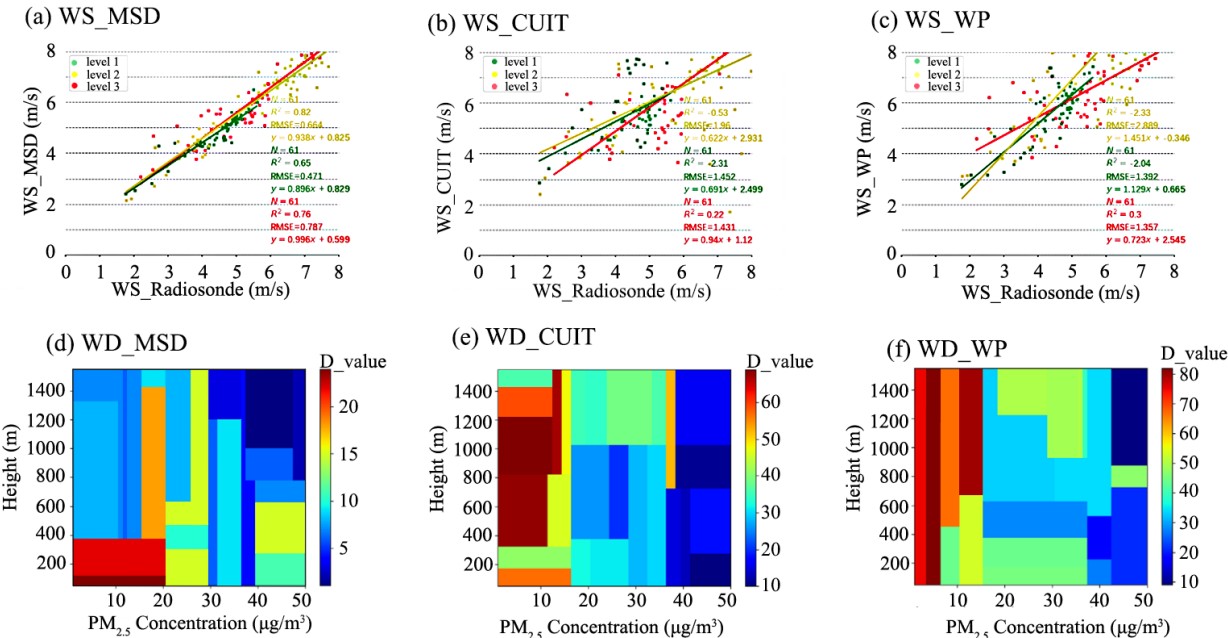

**Fig 5. Comparison of radiosonde data with CUIT, MSD, WP (a-c) WS, and (d-f) WD at different heights and PM2.5 concentrations.**

**Table. 3. Regression parameters of three Lidars and radiosonde under different pollution conditions.**

| Lidar | Level | Fitted curve | N | R2 | RMSE |
|-------|-------|--------------|---|-----|------|
|       | L3    | y=0.938x+0.825 | 61 | 0.82 | 0.66 |
| MSD   | L2    | y=0.896x+0.829 | 61 | 0.65 | 0.47 |
|       | L1    | y=0.996x+0.599 | 61 | 0.76 | 0.79 |
|       | L3    | y=0.622x+2.931 | 61 | 0.24 | 1.96 |
| CUIT  | L2    | y=0.692x+2.499 | 61 | 0.30 | 1.45 |
|       | L1    | y=0.94x+1.12   | 61 | 0.35 | 1.43 |
|       | L3    | y=1.451x+0.346 | 61 | 0.04 | 2.89 |
| WP    | L2    | y=1.129x+0.665 | 61 | 0.17 | 1.39 |
|       | L1    | y=0.723x+2.545 | 61 | 0.32 | 1.36 |





### 3.3 PBLH's impact on Doppler wind Lidars

Aerosol concentrations exhibit a pronounced inverse correlation with PBLH variations. During daytime, convective updrafts enhance PBLH development, which promotes vertical diffusion of aerosols and reduces their near-surface concentrations (Paul and Das, 2022; Su et al., 2018). This section quantifies explicitly the sensitivity of Lidar wind field retrievals to PBLH stratification. Performance evaluations of three Lidar systems (MSD, CUIT, and WP Lidar) against radiosonde measurements were conducted across PBLH and CBH used by the ERA5 datasets. Because of the linear relationship, the Pearson correlation coefficient (PCC) was chosen to represent the correlation with the radiosonde. As shown in Fig. 6, the MSD Lidar exhibited a correlation higher than 0.85 with radiosonde wind speed across all height intervals, demonstrating strong accuracy and insensitivity to PBLH variations. However, its wind direction correlation notably decreased to 0.53 within the 1500–1750 m PBLH stratum, likely attributable to enhanced aerosol-layer complexity at elevated mixing heights and the small samples in this range (N = 41). Outside this interval, wind direction correlations remained robust (>0.70), indicating superior overall performance. The CUIT Lidar showed a wind speed correlation generally above 0.7, its performance was optimal in the 500-750m PBLH range ($\rho_{WS} = 0.92, \rho_{WD} = 0.75$), but its wind speed decreased to 0.6 at a PBLH of 1000-1250m. The wind direction correlation dropped below 0.4 when PBLH exceeded 1500m, reflecting limitations in high-altitude detection. The WP Lidar showed significant deficiencies in wind speed detection, with correlations below 0.72 across all height intervals, and its performance declined notably with increasing PBLH.

This may be attributed to the principle of DWL, which posits that backscattering signals from aerosols play a critical role in wind speed measurement, particularly within the boundary layer and lower troposphere (He et al., 2022b; Li and Yu, 2018; Tan et al., 2019). The performance of Lidar is influenced by the distribution of particles, which is affected by different PBLH levels. At lower altitudes (<750m), all three Lidars demonstrated optimal performance, likely due to stable wind speed and direction, and minimal turbulence within this range, resulting in superior Lidar measurement accuracy. Conversely, the wind direction measurement performance declined substantially at higher altitudes (>1500m).



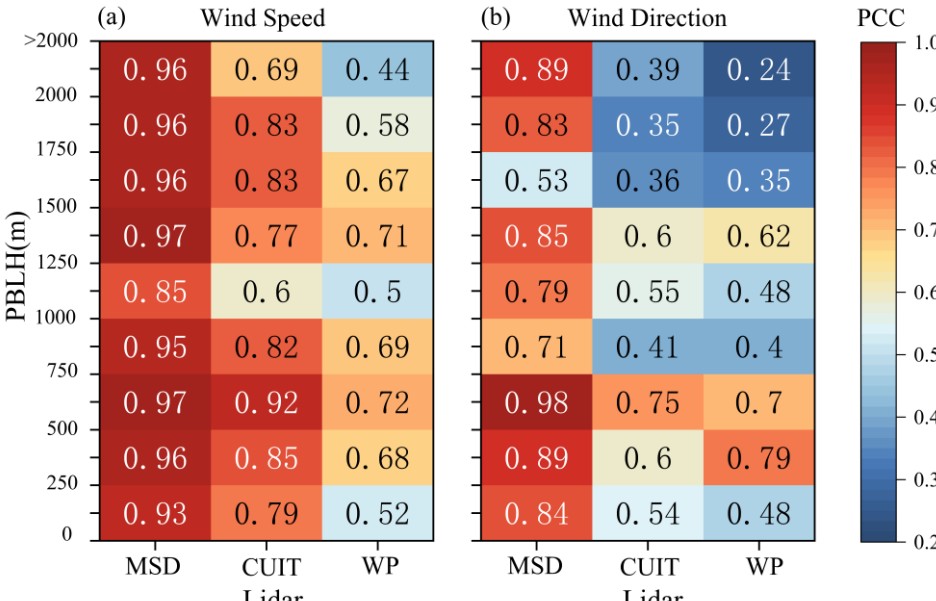

**Fig. 6. Comparison of (a) WS and (b) WD between radiosonde data and CUIT Lidar, MSD Lidar, and WP Lidar at different PBLH;**
**Color bar represents Pearson correlation coefficient.**

To further investigate the influence of PBLH on Lidar performance, the vertical relationship between CBH and PBLH (both
derived from ERA5 data) was introduced to analyze atmospheric impacts on Lidar measurements. As illustrated in Fig. 6, two
distinct PBLH ranges, 1000-1250 m and 500-750 m, were selected to examine contrasting Lidar performance (superior vs.
inferior). The WS and WD performance of three Lidars under varying CBH and PBLH conditions is summarized in Table 4.

When PBLH was elevated (1000-1250 m) with low clouds (CBH < 1000 m), the PCCs for MSD are 0.85 (WS) and 0.93 (WD).
Under shallow PBLH (500-750 m) with higher clouds (CBH > 750 m), MSD exhibited significantly improved PCCs of 0.97
(WS) and 0.98 (WD), maintaining its superior performance. Notably, MSD and CUIT Lidars dominated in WS correlation
(PCCs: 0.85 and 0.59, respectively) under high PBLH conditions (1000-1250 m, CBH < 1000 m), which is similar to Fig. 6.
In this case, the coupling ratio between cloud and PBLH is as high as 90%(Su, 2022), the atmosphere is usually accompanied

by higher relative humidity(Liu, 2019), the turbulent mixing effect in the boundary layer is enhanced, and the vertical
distribution of aerosols becomes complicated, all of which exacerbated Lidar signal interference. Conversely, PBLH was
elevated (500–750 m) with high clouds (CBH > 750 m), WD correlations dominated across all three Lidars (PCCs: 0.98, 0.65,
and 0.59). The decoupling between clouds and the boundary layer fostered a stable vertical structure, confining aerosols and
turbulence predominantly below the PBLH. This stratification minimized cloud-induced signal attenuation, enabling clearer

detection of vertical wind profiles.

**Table. 4. The WS and WD performance of three Lidars under varying CBH and PBLH conditions.**



| Condition | PBLH | CBH | MSD | CUIT | WP |
|-----------|------|-----|-----|------|-----|
| WS | 1000-1250 | >1250 | 0.82 | 0.41 | 0.57 |
| | | <1000 | 0.85 | 0.59 | 0.29 |
| | 500-750 | >750 | 0.97 | 0.48 | 0.64 |
| | | <500 | 0.91 | 0.85 | 0.80 |
| WD | 1000-1250 | >1250 | 0.76 | 0.55 | 0.52 |
| | | <1000 | 0.93 | 0.52 | 0.28 |
| | 500-750 | >750 | 0.98 | 0.65 | 0.59 |
| | | <500 | 0.65 | 0.42 | 0.28 |

**3.4 Analysis of Correction Results of Random Forest Algorithm**

The IF filtering identified additional data gaps in both CUIT and WP Lidar datasets. To maintain temporal continuity, anomalous values were replaced with NaN rather than row deletion. Five interpolation algorithms—CSI, BPNN, GA, k-NN, and RF—were implemented to enhance data reliability. Fig. 7 shows the comparison of the ROC curves of five optimization algorithms. The AUC metrics, accompanied by 95% confidence intervals (CI) derived from bootstrap resampling (n=1,000). Random forest (RF_CUIT and RF_WP) demonstrated superior performance with the AUC of 0.93 (95% CI [0.91–0.94]) and

0.90 (95% CI [0.89–0.91]), underscoring its robustness in modeling non-linear relationships and high-dimensional atmospheric data. This aligns with its inherent capability to handle complex interactions within lidar-derived wind profiles. CSI's inherent locality is characterized by using cubic functions to connect adjacent points (Komsta, 2010). This approach entails global fitting, thereby rendering any alteration in a single data point capable of affecting the entire curve. This heightened sensitivity can make the spline curve more uneven and challenging to manipulate, particularly for functions comprising linear segments

or sudden alterations (Maglevanny and Smolar, 2016). Wind speed may exhibit nonlinear or abrupt variations over time and space under higher altitudes or complex airflow conditions. Cubic spline interpolation struggles to capture such non-smooth dynamic changes effectively, leading to increased interpolation errors. For BPNN, the sufficiency and efficiency of the training set are critical factors influencing generalization (Singh et al., 2023). With limited wind speed data, there is a risk of overfitting or underfitting, which may lead to unstable performance. Due to the inherent randomness of genetic operations, the GA does

not always produce optimal solutions, although it can find suboptimal solutions within a reasonable time (Jurasovic and Kusek, 2010). While GA excels in global optimization, its iterative nature might hinder real-time processing efficiency, a critical factor for operational Lidar systems. The k-NN algorithm is sensitive to local data structures, performing poorly in regions





with sparse observations or high volatility (Gupta et al., 2020). The RF algorithm can handle high-dimensional datasets and capture complex nonlinear relationships effectively (Grimm et al., 2008; Horemans et al., 2020). RF minimizes the risk of

overfitting, resulting in more reliable predictions by aggregating multiple decision trees (Grimm et al., 2008; Horemans et al., 2020; Li et al., 2023). Regarding interpolation, RF can effectively handle missing values, ensuring that the model remains robust and accurate (Xu et al., 2024). The iterative hyperparameter tuning process optimized RF's performance, confirming its suitability for DWL data correction under complex atmospheric conditions.

In summary, all algorithms significantly outperformed the random guess baseline (AUC = 0.5), and the confidence intervals

across all methods are narrow (<0.04 AUC range), confirming their utility and reliability in wind data refinement. recommended. RF is recommended for Lidar applications, prioritizing accuracy due to its high AUC and stable CI. These findings highlight the importance of algorithm selection tailored to specific operational requirements in atmospheric remote sensing.

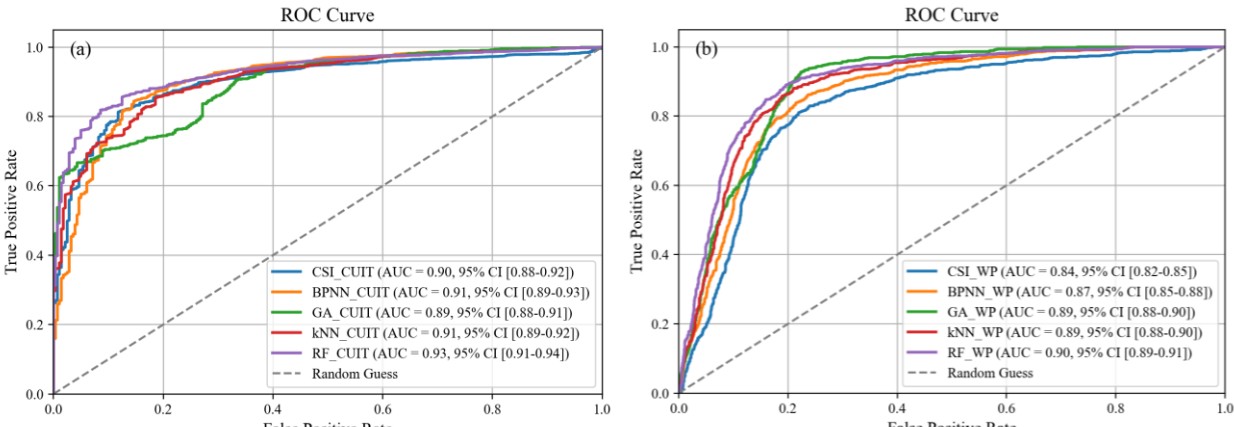


**Fig. 7. The ROC Curves between three Lidars and radiosonde after interpolation using five algorithms.**

To achieve optimal interpolation results, a parameter grid was defined with "mtry" and "ntree". "mtry" represents the number of features considered at each split in the RF, with $mtry \in [1,10]$, and "ntree" represents the number of trees, with $ntree \in [100,500]$. The parameter grid was iteratively traversed in the training function to identify the optimal parameter configuration,

with a fixed random seed ensuring computational reproducibility.

Fig. 8a shows the scatter plot of wind speed data after removing anomalies using the IF and interpolating missing values with RF. Initial CUIT Lidar wind speed data exhibited poor agreement with radiosonde measurements ($R^2 = 0.42$). Following anomaly removal via the IF and RF-based interpolation, correlation improved significantly ($R^2 = 0.65$). The RF interpolation led to a more complete data distribution, with missing values being compensated for.

Fig. 8b shows the distribution of differences between CUIT Lidar and radiosonde data after algorithmic processing. The original CUIT data (green) exhibited a wide distribution with a maximum difference of 34 m/s. The peak of the difference was



not concentrated near 0 m/s. After filtering the data with IF (blue), anomalies were removed, and the differences became more concentrated within -3 to 3 m/s. The IF effectively identifies and outliers, allowing the remaining data to better align with the radiosonde trend. After optimizing and supplementing the data using the RF algorithm (orange), the wind speed data became

closer to the radiosonde data, with the peak difference aligning at 0 m/s. The range of the difference distribution further narrowed, demonstrating a high consistency between the interpolated CUIT Lidar data and the radiosonde data. The orange histogram exhibited significantly superior symmetry and concentration compared to the blue histogram, RF not only repaired missing values but also preserved the global characteristics and trends of the data. In summary, the enhanced peak concentration of the difference distribution validates the applicability and reliability of the RF model in correcting nonlinear

data. These improvements are particularly pronounced in low-altitude regimes (<1 km), where boundary layer turbulence amplifies measurement uncertainties.

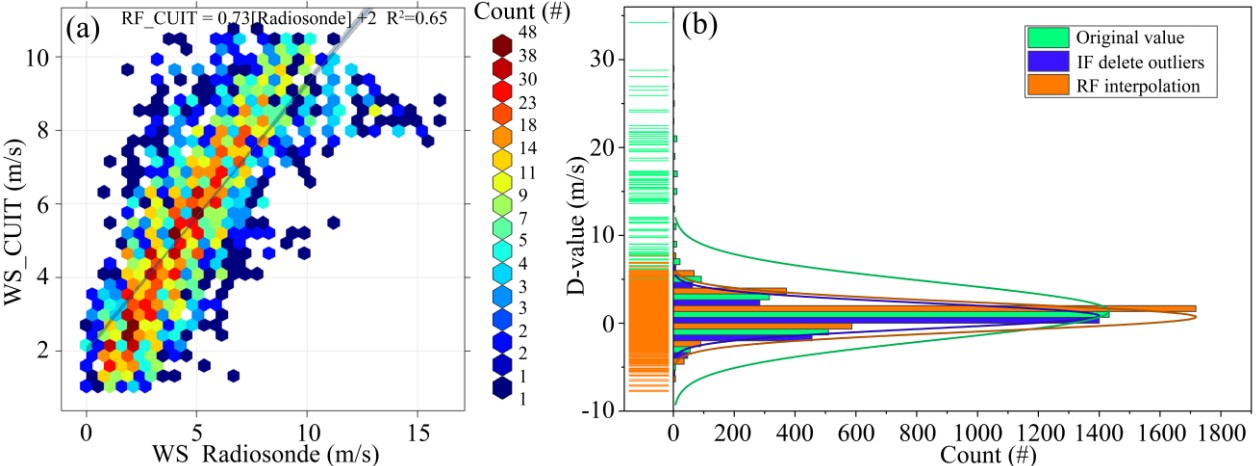

**Fig. 8. (a) WS scatter plot of CUIT Lidar and radiosonde after processing; (b) Comparison distribution map of the difference**
**between CUIT and radiosonde after IF and RF processing.**

### 3.5 Aeolus verification

Cloud cover significantly influences DWL measurement fidelity, as evidenced by comparative analysis with Aeolus satellite products(Guo, 2021). As shown in Fig. 9, comparative analysis of Aeolus satellite products revealed enhanced wind speed retrieval precision under cloudy conditions (Mie-channel $R^2 = 0.90$) compared to clear-sky retrievals (Rayleigh-channel

$R^2 = 0.88$), consistent with Guo's observation of cloud-enhanced signal calibration. This performance differential stems from amplified backscatter signals through cloud-aerosol interactions, underscoring the critical role of atmospheric particulates as natural scattering tracers for optimizing spaceborne wind profiling.

The Aeolus satellite exhibits high consistency with radiosonde data in both channels, indicating the feasibility of using Aeolus for Lidar data validation. A case for a radiosonde observation on June 17, 2021, as shown in Fig. 9a and 9b, indicates that




CUIT Lidar data had a high proportion of missing values, with a missing rate of up to 80%. However, in this case, the application of RF for interpolation led to a substantial enhancement in the congruence between CUIT Lidar and radiosonde data, particularly within the 0.5-1 km altitude range. During the radiosonde observation on June 18 in Fig. 9c and 9d, the IF successfully identified and removed anomalies in the 200-600m range. Following RF interpolation, the correlation between CUIT Lidar and Aeolus satellite data exhibited a substantial enhancement, with $R^2$ reaching 0.83. This outcome signifies

that this method can effectively enhance data quality and accuracy, even in anomalies. Integrating Aeolus validation and RF-based correction establishes a robust framework for enhancing Lidar data reliability. These findings validate the ability of machine learning for complex atmospheric data reconstruction.

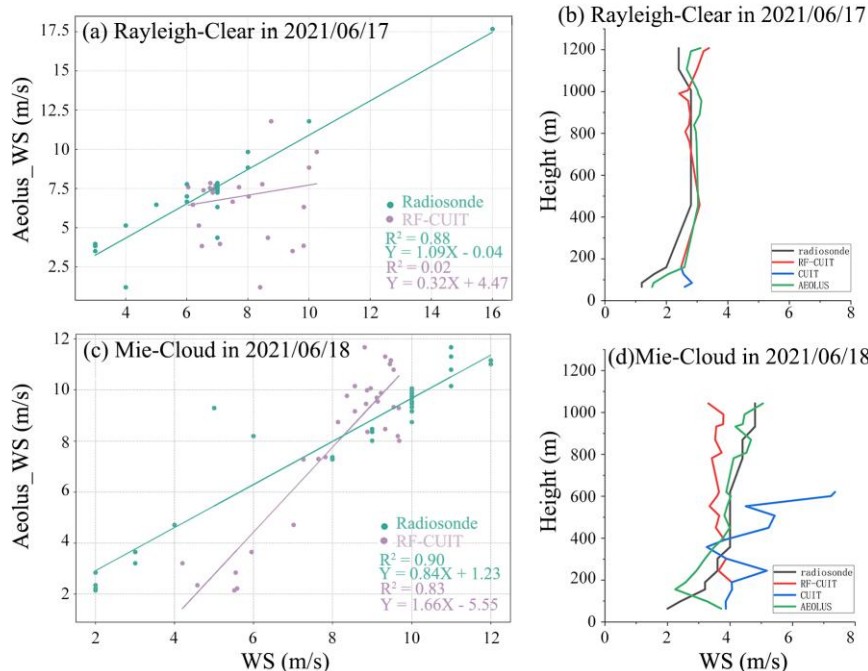

**Fig. 9. The relationship between interpolated CUIT Lidar, radiosonde, and Aeolus satellite data in the (a) Rayleigh and (c) Mie**
**channels; and (b, d) a profile plot from a single radiosonde observation.**

## 3 Conclusion

This study was conducted at the Nanjiao Observatory in Beijing from June 9 to August 31, 2021, using three ground-based DWLs (MSD, CUIT, and WP) and simultaneous radiosonde data to evaluate the performance of the Lidars under different conditions.

The results show that all Lidars demonstrate strong concordance with radiosonde wind speed measurements at the low altitude of 600 m. As altitude increases, the deviations in wind speed and direction from the three Lidars gradually increase. The RMSE



of wind speed for MSD Lidar is 1.11 m/s, 4.45 m/s for CUIT Lidar and 5.15 m/s for WP Lidar. In terms of wind direction, MSD Lidar exhibited the most accurate performance, with an RMSE of 49.83°, CUIT Lidar with an RMSE of 82.88°, and WP Lidar exhibited the most significant deviation with an RMSE of 84.87°. Among the three Lidars, MSD Lidar exhibited the

highest degree of accuracy and correlation in wind speed and wind direction measurements, which is the closest to the radiosonde.

The correlation and accuracy of wind speed measurements from MSD Lidar with radiosonde data were optimal under varying pollution conditions, as evidenced by $R^2$ values of 0.76, 0.65, and 0.82 for L1, L2, and L3 pollution conditions, and RMSE values of 0.79 m/s, 0.47 m/s, and 0.66 m/s, respectively. Additionally, under light pollution conditions with aerosol mass

concentrations of 0-15 µg/m³, MSD Lidar exhibited the highest correlation with radiosonde wind speed, demonstrating its intense sensitivity to aerosol mass concentrations. When the aerosol concentration in the lower atmosphere increases to a certain level (40-50 µg/m³), Lidar can facilitate better signal reception by scattering improvement. Consequently, it is imperative to consider the impact of varying aerosol mass concentrations when detecting low-altitude wind fields to ensure the optimal performance of Lidar instruments.

It has been demonstrated that PBLH has a notable impact on Lidar performance, with the most significant effect observed at PBLH of 1000-1250m, and the optimal performance at lower altitudes (500-750m). MSD and CUIT Lidars dominated in WS correlation (PCCs: 0.85 and 0.59, respectively) under high PBLH conditions (1000-1250 m, CBH < 1000 m). The turbulent mixing effect in the boundary layer is enhanced, and the vertical distribution of aerosols becomes complicated, which exacerbates Lidar signal interference. Conversely, PBLH was elevated (500–750 m) with high clouds (CBH > 750 m), WD

correlations dominated across all three Lidars (PCCs: 0.98, 0.65, and 0.59). The decoupling between clouds and the boundary layer fostered a stable vertical structure. This stratification minimized the cloud-induced signal attenuation.

Five algorithms interpolation (CSI, BPNN, GA, k-NN, and RF) was applied to CUIT and WP Lidar, the RF demonstrated superior performance with the AUC of 0.93 (95% CI [0.91–0.94]) and 0.90 (95% CI [0.89–0.91]) in the ROC curves. And RF-based correction of CUIT enhanced R² from 0.42 to 0.65, bringing it into closer alignment with the radiosonde data. This

outcome underscores the efficacy of the RF correction algorithm, its reliability, and its aptitude for managing high-dimensional and incomplete data.

The cloud cover has a significant impact on the DWL measurement by the comparative analysis with the Aeolus satellite product, the results revealed enhanced wind speed retrieval precision under cloudy conditions (Mie-channel $R^2 = 0.90$) compared to clear-sky retrievals (Rayleigh-channel $R^2 = 0.88$). In the case of severe anomalies, the correlation between

CUIT Lidar and satellite data is significantly enhanced after RF interpolation, and $R^2$ reaches 0.83.

Overall, this study sheds light on the different factors affecting the DWLs of wind speed and wind direction, including different aerosol mass concentrations, PBLH and CBH conditions, Machine learning, and Satellites, and the combination of IF and RF algorithms can effectively improve the quality and accuracy of wind field data for the future research of low-altitude detection.



**Data availability:** (Zhang, 2025)Zhang, Yidan (2025), "Performance Comparison and Wind Speed Correction Algorithm for Three Lidar Wind Profilers", Mendeley Data, V1, doi: 10.17632/s7rjpshdpm.1

**Author contributions:** YZ performed the data collection and analysis, wrote the manuscript, HH performed the data presentation, HW provided the idea and paper revision, FZ performed the supervision, and HS provided the analysis method.

**Competing interests:** The authors declare that they have no conflict of interest.

**Acknowledgments:** We are very grateful to the Meteorological Observation Centre of China Meteorological Administration for the support of this test, and the contributions of teachers and students of CUIT who participated in the task of observation maintenance and support.

**Financial support:** This work was funded by the Chengdu Science and Technology Program (2023-YF09-00013-SN).



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
