# Peer review of "Comparison of the Performance between Three Doppler wind Lidars and a Novel Wind Speed Correction Algorithm"

_EGUsphere, 2025_

## Author Comment (AC1)

Responses to the Reviewer 1 comments

(comments in black, responses in red)

General comments:

The manuscript presents a comprehensive comparative analysis of three Doppler wind lidars (DWLs) under varying environmental and technical conditions, alongside a novel machine learning-based correction algorithm. The systematic evaluation of three DWLs under diverse environmental conditions (aerosol concentration, planetary boundary layer height (PBLH), cloud base height (CBH)) provides actionable insights for instrument deployment and data interpretation. The discussion of aerosol concentration as a critical factor for low-altitude wind profiling is well-supported and relevant for urban air quality and aviation studies. The novel integration of IF and RF algorithms to correct lidar data is a notable contribution, particularly the validation against Aeolus satellite retrievals. However, several aspects require clarification, methodological justification, or refinement to strengthen the manuscript's impact. Below are specific comments and suggestions.

The language is generally good, technical, and appropriate for a scientific preprint. It effectively communicates complex research in atmospheric science and remote sensing. The logic is sound and well-structured, following a standard scientific paper format. The fluency is mostly good, though there are some minor grammatical issues and slightly awkward phrasing that could be improved for enhanced clarity and readability.

Dear Reviewer,

Thank you for your thoughtful and constructive feedback on our manuscript titled "Comparison of the Performance between Three Doppler wind Lidars and a Novel Wind Speed Correction Algorithm." We sincerely appreciate the time and expertise you dedicated to reviewing our work, especially your insightful suggestions regarding the Fengyun satellite data, aerosol concentration considerations, radar-related analyses, and the numerous refinements to the manuscript's clarity and technical rigor. Your comment has been invaluable in strengthening the paper. We have carefully addressed each point raised — revising sections for precision, enhancing methodological explanations, clarifying details, and incorporating additional context where needed. The manuscript has been significantly improved as a direct result of your meticulous review.

The following are the detailed responses:

Major comments:

Ambiguity in the finding on aerosol impact: The paper repeatedly states that higher $PM_{2.5}$ concentrations (L3: 35-50 $\mu g/m^3$) lead to improved Lidar performance (e.g., "Particle mass concentration loading has positive correlation on the Lidar performance", "MSD Lidar exhibited the highest wind speed correlation($R^2$= 0.82)... when $PM_{2.5}$ ranges from 35 to 50 μg/m³", "the wind direction accuracy... is enhanced with the increase of aerosol concentration", "indicating that particle loading is the critical factor").

My concerns are follows: This finding appears counter-intuitive and contradicts fundamental Lidar physics and established literature. While aerosols are necessary as scattering targets for coherent DWL, excessively high aerosol concentrations (like 35-50 μg/m³, typical of polluted conditions) cause significant signal attenuation. This attenuation should degrade signal-to-noise ratio (SNR) and measurement accuracy, especially at higher altitudes, not improve it. The observed correlation

improvement under L3 conditions needs a much more robust and physically sound explanation.

Reply: "Aerosols serve as essential scattering targets for coherent Doppler lidar, but excessively high aerosol concentrations induce significant signal attenuation. This is well-established in the literature. According to China's ambient air quality classification standards (based on $PM_{2.5}$ concentrations): Excellent: 0–35 µg/m³

Good: 35–75 µg/m³

Light Pollution: 75–115 µg/m³

Moderate Pollution: 115–150 µg/m³

Heavy Pollution: 150–250 µg/m³

Severe Pollution: ≥250 µg/m³

Wu et al. [1]mentioned that by combining MODIS satellite data with ground observations, the relationship between $PM_{2.5}$ concentration and aerosol optical thickness (AOD) was quantified, and it was pointed out that the inversion error of lidar increased under high $PM_{2.5}$ concentration. When $PM_{2.5} > 100$ µg/m³, AOD > 0.8, and the attenuation rate of the radar signal increases by more than 40%.

In our study, the experiments were conducted in Beijing during summer (June-August), a period characterized by naturally lower aerosol loading. To achieve methodological clarity, we stratified the data into three levels:

L1 ($PM_{2.5}$ = 0–15 µg/m³)

L2 ($PM_{2.5}$ = 15–35 µg/m³)

L3 ($PM_{2.5}$ = 35–50 µg/m³)

Notably, our results indicate that lidar performance exhibits enhanced sensitivity at the L3 level (35–50 µg/m³), which aligns with the threshold where aerosol scattering begins to optimize signal-to-noise ratios without triggering severe attenuation. We acknowledge that the phrasing in the original manuscript could be misinterpreted and we will refine the description to explicitly state:

"The observed performance improvement at L3 concentrations reflects that the Lidar requires a certain amount of aerosol backscattering rather than implying superior functionality under polluted conditions."

This clarification will be incorporated into the revised text to eliminate ambiguity.

Insufficient discussion of vertical resolution mismatch in Aeolus validation: The validation of the RF-corrected CUIT Lidar data against Aeolus satellite data (showing $R^2$=0.83) is presented as a key result demonstrating the effectiveness of the correction algorithm. However, a critical methodological limitation is glossed over.

My concern is the Aeolus L2B product has a relative coarse vertical resolution (0.25 km to 2 km bins, Table 1), while the ground-based Lidars (especially CUIT/MSD with 30m/50m/60m resolution) provide high-resolution profiles. Comparing a point measurement (radiosonde) or a high-resolution profile (Lidar) to a vertically averaged satellite product (Aeolus) is inherently problematic. The reported $R^2$ value might be significantly influenced by this mismatch rather than solely reflecting the accuracy of the RF correction or the Lidar itself. Averaging the high-resolution Lidar data to match Aeolus bins before calculating $R^2$ is essential for a fair comparison.

My recommandations are 1. Explicitly state the vertical resolution mismatch as a significant limitation in the Aeolus validation section. 2. Discuss how the coarse Aeolus resolution might impact the validation result (e.g., smoothing out small-scale features the Lidar might resolve,

potentially inflating agreement). Acknowledge that the R²=0.83 reflects agreement on the scale resolvable by Aeolus, not necessarily the full resolution of the Lidar.

Reply: We acknowledge that the vertical resolution of Aeolus constitutes a significant constraint in our analysis.

The satellite-Lidar comparison in our manuscript references the method proposed by Guo et al.[2]. The Aeolus Level 2B wind products represent averages over specific vertical bins (each bin spanning 0.25–2 km in height), while ground-based instruments achieve resolutions of 30 m/50 m/60 m. Directly comparing single-point measurements with vertically averaged layers would introduce bias.

Preprocessing Steps:

Step 1: Partition the high-resolution ground-based lidar data according to Aeolus's vertical bin boundaries.

Step 2: Average the ground-based data within each bin to generate vertical-layer-averaged wind fields corresponding to Aeolus.

Step 3: Project the averaged wind fields onto Aeolus's line-of-sight (HLOS) direction for comparison.

Data filtering was performed based on the Aeolus L2B algorithm theoretical documentation:

Mie-cloudy products are less affected by aerosols; wind speed data < 5 m/s were retained.

Rayleigh products require filtering of high-error samples; wind speed data < 7 m/s were retained.

Through vertical averaging and strict matching, data of differing resolutions can be effectively compared. However, Aeolus represents regional averages, while ground-based instruments represent point measurements, resulting in partial scale discrepancies. As noted in Section 4.2 of Guo et al.[2], "Aeolus achieves sufficient accuracy (RMSE < 3 m/s) below 5 km for data assimilation in NWP models, but requires careful quality control in moist atmospheres." Satellites and radars exhibit optimal consistency (minimal deviation) within the 1–5 km altitude range. Above 5 km, deviations increase due to attenuated atmospheric signals in thin air.

The accuracy of the Aeolus observations in low-altitude not only from the presence of tracer aerosols but also from its intrinsically higher resolution. Our validation approach evaluates Aeolus performance against the satellite's resolution capabilities rather than the full potential resolution of ground-based lidar systems.

The RMSE values for wind speed and direction (e.g., MSD: 1.11 m/s, CUIT: 4.45 m/s) are presented without explicit context. Are these values aggregated across all altitudes (<2 km) or stratified by specific height ranges? Clarify whether these RMSEs are altitude-dependent.

Reply:

L. 10: mentioned that the altitude is below 2 km.

L. 13: The RMSE results (e.g., MSD: 1.11 m/s, CUIT: 4.45 m/s) are the first research results of the manuscript, we will correct the description: "Within the research height range, comparison of results shows the root mean square errors (RMSE) for wind speed were 1.11 m/s, 4.45 m/s, and 5.15 m/s...". We will check the manuscript whether there is any ambiguous expression like this.

The "high-dimensional and incomplete datasets" processed by the IF-RF algorithm require clarification. Specify the input features (e.g., altitude, PM5, PBLH) and the proportion of missing

data imputed.

Reply: The CUIT data is complete without any omission. The one with the most missing data is CUIT Lidar, where 426 out of a total of 2,885 data points are missing, the missing rate reaches 14.8%. The second one is WP Lidar, there are 46 missing data points, accounting for 1.6% of the total 2,885 points.

We will supplement the quantitative analysis in Section 2.2.

The $R2 = 0.83$ between RF-corrected CUIT and Aeolus satellite data is promising, but the manuscript does not discuss potential biases in satellite retrievals (e.g., Aeolus's own uncertainties under cloudy vs. clear-sky conditions). A brief comparison of satellite and radiosonde error profiles would strengthen this section.

Reply: The Aeolus L2B wind product, derived from ALADIN (Atmospheric LAser Doppler INstrument)—the world's first space-borne ultraviolet Doppler wind lidar—provides horizontal line-of-sight (HLOS) wind components. Thick clouds or high-concentration aerosol layers strongly attenuate the laser signal, preventing penetration into atmospheric layers beneath cloud systems. Even when clouds are detected, Rayleigh signals from clear-sky regions below are attenuated, leading to increased uncertainty in low-level clear-sky data. As reported by Rennie et al. [3] and Lux et al. [4], data acquisition rates for lower atmospheric layers significantly decrease under multilayered clouds or optically thick cloud systems.

Mie winds exhibit minimal systematic bias in regions with strong scatterers (typically clouds/aerosols), though random errors vary with signal strength. Rayleigh winds show small biases and random errors in the clear-sky free troposphere but face increased uncertainty in cloud-affected regions or the clear-sky boundary layer. Within cloud layers, Rayleigh channel signals are heavily scattered and absorbed by cloud particles, necessitating reliance on the Mie channel.

Aeolus' strength lies in its global coverage, whereas its weaknesses include vertical resolution, cloud-penetration capability, and high sensitivity to clouds. Radiosondes remain an unparalleled reference benchmark, especially for validating Aeolus under cloudy conditions—despite their spatial representativity limitations. While both datasets perform comparably in the middle-to-upper troposphere under clear skies, radiosondes deliver more reliable, higher-resolution profiles in the lower atmosphere and complex cloud regimes.

The manuscript needs to improve writing quality.

Reply: We have revised the manuscript to make the paragraphs more fluent.

Minor comments:

Pay attention to the use of "the". Like abstract: "Lidars data comparison focus on the low altitudes".

Reply: We will carefully examine these statements to confirm all the expressions of "the".

Coordinates need a degree sign and a space when naming the direction. e.g., (39.80° N, 116.32°E).

Reply: We have modified the direction in the manuscript, in accordance with the submission requirements on the official website.

..like m/s and m s-1...should be serious in writing scientific paper.

Reply: According to the "Figures & tables" section in the submission requirements on the official website, we have revised it to exponentially units (m s-1) in the manuscript.

Check the numbers of section headings. Results and discussions should be 3, and. 1.3.1 Isolation tree should be 2.2.1⋯.

Reply: We didn't carefully check the subheadings. We have modified "1.3.1 Isolation tree" to "2.2.1 Isolation tree ". and "1.3.2 Anomaly Detection" has been modified to "2.2.2 Anomaly Detection ".

A range of numbers should be specified as "a to b" or "a...b"(e.g., L1 (PM5 = 0–15 µg/m³), L2 (PM2.5 = 15–35 µg/m³), and L3 (PM2.5 = 35–50 µg/m³)).

Reply: In the requirements for manuscript submission, ranges need an endash and no spaces between start and end (e.g. 1–10, Jan–Feb). We will examine the full text and make corrections.

The statement "the measurement accuracy decreases with altitude increase (<2 km)" is redundant. Rephrase to "measurement accuracy decreases with increasing altitude (up to 2 km)."

Reply: L13: "the measurement accuracy decreases with the altitude increase (<2 km)" has been modified to "measurement accuracy decreases with increasing altitude (up to 2 km).".

The phrase "particle mass concentration loading has positive correlation on the Lidar performance" is ambiguous. Specify whether higher aerosol concentrations improve or degrade

Reply: L13: Statement has been changed to "Within the range of 0-50µg/m³, the higher aerosol concentration can improve the Lidar accuracy".

The "optimal performance at lower altitudes (500–750 m)" conflicts with the earlier statement that accuracy decreases with altitude. Clarify whether "optimal" refers to relative performance across lidars or absolute performance.

Reply: The performance of the radar is relatively optimal at lower altitudes (500-750 meters), but in terms of the overall height, the Lidar accuracy decreases with increasing altitude (<2km). Paragraph has changed to "Particularly, the Lidar performance is relatively optimal when the PBLH is 500-750m."

MSD=0.97

Summary: "MSD Lidar exhibited the most accurate performance... which is the closest to the radiosonde" → "MSD Lidar exhibited the highest accuracy... closest to radiosonde measurements."

Reply: L399: The statement has been revised to "MSD Lidar exhibited the highest accuracy in wind speed and wind direction measurements, closest to radiosonde measurements.".

Avoid passive voice where possible (e.g., "It has been demonstrated that PBLH..." → "PBLH significantly influences lidar performance...").

Reply: L410: The statement has been revised to "PBLH significantly influences lidar performance…".

"the accuracy of the three ground-based Lidars is evaluated" -> "...Lidars are evaluated".

Reply: L85: The statement "the accuracy of the three ground-based Lidars is evaluated against radiosonde data as the reference standard." has been revised to "Lidars are evaluated against radiosonde data as the reference standard".

"Lidars data comparison focus" -> "Lidar data comparison focuses".

Reply: L10: The statement "Lidars data comparison focus on the low altitudes…" has been revised to "Lidar data comparison focuses on the low altitudes…".

"Comparison of results shows the root mean square errors(RMSE) for wind speed were..." -> "Comparisons show the root mean square errors (RMSE) for wind speed were..."

Reply: L11: The statement "Comparison of results shows the root mean square errors (RMSE) for wind speed were..." has been revised to "Comparisons show the root mean square errors (RMSE)

for wind speed were...".

"the wind direction accuracy observed by the three Lidars is enhanced" -> "...observed with the three Lidars is enhanced".

Reply: L15: The statement "and the wind direction accuracy observed by the three Lidars is enhanced with…" has been revised to "and the wind direction accuracy observed with the three Lidars is enhanced with…".

"Machine learning was used to remove anomalies and complement the missing values" -> "...to remove anomalies and impute missing values"

Reply: L20: The statement "Machine learning was used to remove anomalies and complement the missing values…" has been revised to "Machine learning was used to remove anomalies and impute missing values…".

"the random forest(RF) demonstrated superior performance with the Area Under the Curve(AUC) of 0.93(CUIT) and 0.90(WP)" -> "...superior performance, with AUC values of 0.93 (CUIT) and 0.90 (WP)".

Reply: L21: The statement "the random forest (RF) demonstrated superior performance with the Area Under the Curve (AUC) of 0.93(CUIT) and 0.90(WP)…" has been revised to "the random forest (RF) demonstrated superior performance, with the Area Under the Curve (AUC) of 0.93(CUIT) and 0.90(WP)…".

"RF-based correction of CUIT enhanced R 2from 0.42 to 0.65." -> "RF-based correction of CUIT data enhanced the $R^2$ value from 0.42 to 0.65.".

Reply: L23: The statement "RF-based correction of CUIT enhanced $R^2$ from 0.42 to 0.65." has been revised to "RF-based correction of CUIT data enhanced the $R^2$ value from 0.42 to 0.65.".

"The R 2between RF-based CUIT and Aeolus satellite is shown as 0.83" -> "The $R^2$ between the RF-corrected CUIT data and Aeolus satellite data was 0.83"

Reply: L20: The statement "The $R^2$ between RF-based CUIT and Aeolus satellite is shown as 0.83," has been revised to "The $R^2$ between the RF-corrected CUIT data and Aeolus satellite data was 0.83".

"Fig.1. Location and instruments..." -> "Figure 1. Location of the Nanjiao Observatory and instruments deployed during the campaign."

Reply: L20: The statement "Fig. 1. Location and instruments of the radiosonde station." has been revised to "Figure 1. Location of the Nanjiao Observatory and instruments deployed during the campaign.".

"Author contributions: YZ performed the data collection and analysis, wrote the manuscript, HH performed the data presentation..." ("Performed the data presentation" is vague).

Reply: L432: The statement "HH and YL performed the data presentation," has been revised to "HH and YL performed the method improvement and data visualization,".

References:

1.  Wu, J., et al., *VIIRS-based remote sensing estimation of ground-level PM2.5 concentrations in Beijing–Tianjin–Hebei: A spatiotemporal statistical model.* Remote Sensing of Environment, 2016. **184**: p. 316-328.

2.  Guo, J., et al., *Technical note: First comparison of wind observations from ESA's satellite mission Aeolus and ground-based radar wind profiler network of China.* Atmospheric Chemistry and Physics, 2021. **21**(4): p. 2945-2958.

3.  Rennie, M.P., et al., *The impact of Aeolus wind retrievals on ECMWF global weather forecasts.* Quarterly Journal of the Royal Meteorological Society, 2021. **147**(740): p. 3555-3586.

4.  Witschas, B., et al., *First validation of Aeolus wind observations by airborne Doppler wind lidar measurements.* Atmospheric Measurement Techniques, 2020. **13**(5): p. 2381-2396.

---

## Author Comment (AC2)

Responses to the Reviewer 2 comments

(comments in black, responses in red)

General comments:

The manuscript demonstrates a rigorous comparative analysis of three DWLs under different atmospheric conditions and integrates novel ML techniques to address data anomalies and gaps. The findings on PBLH-dependent lidar performance have direct implications for urban air quality studies and wind energy applications, where boundary layer dynamics are paramount. The proposed IF-RF pipeline offers a scalable solution for operational lidar networks, particularly in polluted regions with high aerosol variability. By evaluating lidar performance against radiosonde and satellite datasets, the study provides actionable insights into the interplay between aerosol dynamics, boundary layer processes, and instrument accuracy. While the work is methodologically robust, several areas require refinement to enhance clarity, statistical rigor, and physical interpretability. I consider this manuscript adequate for publication in Atmospheric Measurement Techniques once my comments are addressed.

Dear Reviewer,

We extend our sincere gratitude for your thorough and insightful review of our manuscript, "Comparison of the Performance between Three Doppler wind Lidars and a Novel Wind Speed Correction Algorithm." Your expertise has been invaluable, particularly regarding: the scarcity of wind field data in the PBLH layer, impacts of aerosol hygroscopic growth and cloud microphysics on radar signal attenuation, temporal resolution limitations of ERA5 reanalysis data, the influence of aerosol concentration on radar performance and so on. After the revision the logic structure and scientific statement has been significantly improved. We have carefully addressed each point raised—revising sections for precision, enhancing methodological explanations, clarifying details where needed. The manuscript has been significantly improved as a direct result of your meticulous review.

The following are the detailed responses:

Major comments:

1.The manuscript should explicitly acknowledge the significant disparity in sample sizes across the analyzed PBLH strata, particularly the notably small sample (N = 41) within the 1500–1750 m stratum. This limited representation inherently reflects the rarity of PBLH events in the study region, as discussed in Section 3.3 ('PBLH's impact on Doppler wind Lidars'). Consequently, any statistical interpretations or conclusions drawn specifically from this stratum require considerable caution and should be clearly qualified. We recommend discussing the limited statistical power and the inherent challenge of capturing sufficient events.

Reply: In section "3.3 PBLH's impact on Doppler wind Lidars," line L280 state that within the PBLH range of 1500–1750 m, the Lidar data samples are relatively few (N=41). However, for the MSD Lidar, wind speed was unaffected and even showed a high correlation (0.96); it was the particularly poor performance in wind direction (0.53) that stood out.

The following figure shows the correlation analysis of wind speed and directions between radiosonde and three Lidars at the PBLH of 1500-1750 m. The selected part shows that the wind direction correlation coefficient between radiosonde and MSD is 0.53. Although there is less data

at this range, it can be seen that the strong correlation of wind speed, while the weak correlation of wind direction due to the large deviation.

[Figure]

Yamartinoz proposed [1] that even when wind speed errors are small, wind direction errors can be significant, especially under low wind speeds or strong turbulence conditions. Wind direction is more sensitive than wind speed to localized, instantaneous small-scale turbulent disturbances or slight variations in airflow direction. A brief shift in airflow or a vortex can significantly alter instantaneous wind direction measurements but may have a smaller impact on average wind speed. Within the high PBLH layer (near the boundary layer top), turbulent intermittency, entrainment processes, and residual layer influences are typically stronger, potentially leading to more unstable airflow direction.

The 1500–1750 m range represents a relatively high boundary layer height, which may correspond to a thicker aerosol layer with potentially more heterogeneous vertical structure (multi-layered, concentration gradients, particle properties). If these aerosol populations carry slightly different local wind information (e.g., due to slight wind shear) or if low aerosol concentration reduces the signal-to-noise ratio, the synthesized wind direction vector will contain greater uncertainty.

Lothon, M. et al.[2] demonstrated that complex turbulent structures and aerosol distributions, which complicate wind field (especially direction) measurements, are typically present at the boundary layer top (high PBLH regions).

We will revise the text in the manuscript to: "However, its wind direction correlation notably decreased to 0.53 within the 1500–1750 m PBLH range, likely attributable to enhanced aerosol-layer complexity at elevated mixing heights and the small samples in this range (N = 41). Although the sample size within this PBLH is relatively less, wind speed was unaffected; only the poor performance in wind direction was particularly prominent. This may be due to complex turbulent structures and aerosol distributions leading to wind direction instability."

2.The study links PBLH and CBH to DWL performance but does not address how cloud

microphysical properties (e.g., hydrometeor phase, particle size distributions, liquid/ice water content) modulate signal attenuation. High humidity under coupled PBLH-CBH conditions likely promotes hygroscopic growth, altering aerosol scattering properties. This microphysical-aerosol interaction mechanism is especially critical to consider in complex, polluted urban environments like Beijing, where aerosol loading and composition are highly variable and can profoundly influence lidar performance. We strongly recommend the authors discuss this potential mechanism and its implications for their findings, or explicitly acknowledge this limitation for future work.

Reply: The coupling ratio between cloud and PBLH reaches to 90% under high PBLH conditions. Such regimes feature elevated relative humidity, intensified turbulent mixing, and complex aerosol stratification – all exacerbating lidar signal interference. We acknowledge that the current lidar performance evaluation (particularly in high PBLH-CBH coupled regimes) may be subject to potential interference from aerosol-cloud microphysical interactions. Under high PBLH-CBH coupling (e.g., PBLH > 1000 m and CBH < 1000 m), elevated cloud liquid water content and aerosol hygroscopic growth may interfere with wind vector retrieval accuracy by altering backscattering profiles. Notably, during our summer campaign in Beijing, cloud bases were typically elevated, resulting in infrequent occurrences of PBLH > CBH conditions. This observational constraint limited our ability to systematically investigate hygroscopic growth processes under strong PBLH-CBH coupling scenarios.

3.Ambiguity in High-Aerosol Performance: The assertion that peak lidar accuracy occurs at PM2.5=35-50 $\mu g/m^3$ (L3) contradicts conventional lidar attenuation models. The conclusion that higher PM2.5 concentrations (35-50 $\mu g/m^3$) enhance lidar performance (e.g., Sec. 3.2, Fig. 5) merits further discussion. Given that dense aerosol layers are known to cause signal attenuation (reducing penetration depth and SNR), the observed performance improvement under polluted conditions (L3) appears paradoxical. We recommend expanding the physical interpretation to address how attenuation effects were overcome in this study.

(a) Lidar fundamentals: High aerosol loads increase attenuation, degrading SNR.

Reply:

While aerosols are crucial scattering targets for coherent Doppler lidar, their excessive concentration leads to substantial signal attenuation, a well-documented phenomenon. China's ambient air quality standards categorize conditions based on $PM_{2.5}$ levels as follows:

| Air quality grade | $PM_{2.5}$ levels |
|---|---|
| Excellent | 0–35 $\mu g/m^3$ |
| Good | 35–75 $\mu g/m^3$ |
| Light Pollution | 75–115 $\mu g/m^3$ |
| Moderate Pollution | 115–150 $\mu g/m^3$ |
| Heavy Pollution | 150–250 $\mu g/m^3$ |
| Severe Pollution | ≥250 $\mu g/m^3$ |

Wu et al. [3]mentioned that by combining MODIS satellite data with ground observations, the relationship between $PM_{2.5}$ concentration and aerosol optical thickness (AOD) was quantified, and it was pointed out that the inversion error of lidar increased under high $PM_{2.5}$ concentration. When $PM_{2.5} > 100$ $\mu g/m^3$, AOD > 0.8, and the attenuation rate of the radar signal increases by more than 40%.

Our study, conducted during the summer months (June-August) in Beijing—a period characterized by naturally lower aerosol burdens—stratified data into three levels for methodological clarity:

L1: PM2.5 = 0–15 μg/m³

L2: PM2.5 = 15–35 μg/m³

L3: PM2.5 = 35–50 μg/m³

Notably, our results show enhanced lidar performance sensitivity at the L3 level (35–50 μg/m³). This aligns with the concentration threshold where aerosol scattering optimizes the signal-to-noise ratio before triggering severe attenuation. We recognize that the original manuscript's phrasing was potentially ambiguous. To prevent misinterpretation, we will revise the manuscript to explicitly state: " The observed performance improvement at L3 concentrations reflects that the Lidar requires a certain amount of aerosol backscattering rather than implying superior functionality under polluted conditions."

4.It is necessary to discuss the limitations of the temporal resolution of ERA5 reanalyzed data. Analyzing ERA5 data's accuracy is not as precise as the radiosonde, please discuss the issue of time resolution.

Reply: The time resolution of the PBLH and CBH of the ERA5 reanalysis data is one hour. During the research period of this paper, the radiosonde balloon was released at 7:15 a.m. and 7:15 p.m. every day. The three Lidars matched the time and altitude through the sliding window method, and the data of ERA5 also matched exactly.

Minor comments:

1.Pay attention to the use of plural, like DWLs and DWL⋯

Reply: We will carefully examine these statements to confirm all the expressions of plural.

2.Line 12, Units must be written exponentially (e.g. W m–2).

Reply: According to the "Figures & tables" section in the submission requirements on the official website, we have revised it to exponentially units (m s-1) in the manuscript.

3.Line 75-77, please add references.

Reply: We have added references [4] to the manuscript.

4.Line 77-78, please check whether the reference is correct.

Reply: we have revised it to "Siying et al. examined the seasonal variation in Aeolus satellite detection performance in China by combining ERA5 and radiosonde data, concluding that the satellite's performance is influenced by seasonal factors (Siying et al., 2021)".

5.Line 92, Coordinates need a degree sign and a space when naming the direction (e.g. 30° N).

Reply: We have modified the direction in the manuscript, in accordance with the submission requirements on the official website. (e.g. 39.80° N)

6.Line 96, Common abbreviations to be applied: hour as h (not hr), kilometre as km, metre as m.

Reply: According to the "Figures & tables" section in the submission requirements on the official website, we have revised it to "h", "km", "m".

7.Line 145, what is the IF

Reply: We have revised the manuscript to "The isolation forest (IF) model…"

8.Results and discussions should be 3, and. 1.3.1 Isolation tree should be 2.2.1⋯.

Reply: We didn't carefully check the subheadings. We have modified "1.3.1 Isolation tree" to "2.2.1 Isolation tree ". and "1.3.2 Anomaly Detection" has been modified to "2.2.2 Anomaly Detection ".

9.Line 317, what is ROC?

Reply: We have revised the manuscript to "the comparison of the Receiver Operating Characteristic

(ROC) curves…”.

10.Line 339, what is AUC?

Reply: We have revised the manuscript L318 to “The Area Under the Curve (AUC) metrics…”.

11.Line 382, ranges need an endash and no spaces between start and end (e.g. 1–10, Jan–Feb).

Reply: In the requirements for manuscript submission, ranges need an endash and no spaces between start and end (e.g. 1–10, Jan–Feb). We will examine the full text and make corrections.

12.Avoids accusatory language (e.g., "contradicts" → "appears counter to," "warrants deeper analysis")

Reply: We will carefully examine these statements to confirm all the expressions.

13.Line 419, “RF-based correction of CUIT enhanced $R^2$ from 0.42 to 0.65” → “RF correction significantly enhanced $R^2$ from 0.42 to 0.65”.

Reply: L419: The statement “RF-based correction of CUIT enhanced $R^2$ from 0.42 to 0.65” has been revised to “RF correction significantly enhanced $R^2$ from 0.42 to 0.65”.

14.Simplify conclusions and avoid redundant expressions.

Reply: We will carefully examine the manuscript to avoid redundant expressions and make the article more coherent.

References:

1. YAMARTINO, *A COMPARISON OF SEVERAL SINGLE-PASS ESTIMATORS OF THE STANDARD-DEVIATION OF WIND DIRECTION.* JOURNAL OF CLIMATE AND APPLIED METEOROLOGY, 1984. **23**(9).

2. Lothon, *Doppler Lidar Measurements of Vertical Velocity Spectra in the Convective Planetary Boundary Layer.* BOUNDARY-LAYER METEOROLOGY, 2009. **132**(2).

3. Wu, J., et al., *VIIRS-based remote sensing estimation of ground-level PM2.5 concentrations in Beijing–Tianjin–Hebei: A spatiotemporal statistical model.* Remote Sensing of Environment, 2016. **184**: p. 316-328.

4. Guo, J., et al., *Technical note: First comparison of wind observations from ESA's satellite mission Aeolus and ground-based radar wind profiler network of China.* Atmospheric Chemistry and Physics, 2021. **21**(4): p. 2945-2958.